# FINDE: Neural Differential Equations for Finding and Preserving Invariant Quantities

## Abstract

Neural networks have shown promise for modeling dynamical systems from data. Recent models, such as Hamiltonian neural networks, have been designed to ensure known geometric structures of target systems and have shown excellent modeling accuracy. However, in most situations where neural networks learn unknown systems, their underlying structures are also unknown. Even in such cases, one can expect that target systems are associated with first integrals (a.k.a. invariant quantities), which are quantities remaining unchanged over time. First integrals come from the conservation laws of system energy, momentum, and mass, from constraints on states, and from other features of governing equations. By leveraging projection methods and discrete gradient methods, we propose *first integral-preserving neural differential equations (FINDE)*. The proposed FINDE finds and preserves first integrals from data, even in the absence of prior knowledge about the underlying structures. Experimental results demonstrate that the proposed FINDE is able to predict future states of given systems much longer and find various quantities consistent with well-known first integrals of the systems in a unified manner.

## 1 Introduction

Although neural networks have achieved remarkable results in image and natural language processing [17, 28], they have also been actively investigated for modeling dynamical systems [41]. Target systems include the chemical dynamics to accelerate computer simulations [46], the climate dynamics for climate change prediction and weather forecasting [47, 52], and the physical dynamics of vehicles and robots for optimal control [41]. Their history dates back to at least the 1990s, and many approaches have been proposed so far (see [7, 12, 35, 40, 49, 55] for example). Recently, neural ordinary differential equation (NODE) has redefined neural networks for continuous-time dynamics [8]. A target system is described by an ordinary differential equation (ODE) $\frac{\mathrm{d}}{\mathrm{d}t}\boldsymbol{u} = f(t, \boldsymbol{u})$, where $\boldsymbol{u}$ denotes the system state. Then, a NODE replaces the vector field $f$ with a neural network and employs a numerical integrator to obtain a solution $\boldsymbol{u}(t)$.

Most real-world systems are associated with *first integrals* (a.k.a. invariant quantities), which are quantities remaining unchanged over time [27]. If a system has a first integral $V(\boldsymbol{u})$, the solution $\boldsymbol{u}(t)$ for the initial condition $\boldsymbol{u}(0)$ remains at a contour line $V(\boldsymbol{u}(t)) = V(\boldsymbol{u}(0))$ over time. Many previous studies have attempted to learn a target system accurately by incorporating prior knowledge about first integrals. Greydanus et al. [26] proposed Hamiltonian neural network (HNN), which employs a neural network to approximate Hamilton's equation, thereby conserving the system energy called the Hamiltonian. Finzi et al. [19] proposed neural network architectures that conserve linear and angular momenta by utilizing the graph structure. Finzi et al. [20] also extended HNN to a system with holonomic constraints, which lead to first integrals such as a pendulum length. Matsubara et al.

Table 1: Comparison between Related Studies on Preservation of First Integrals.

| | energy | momentum | mass | constraint | learning invariants | exact conservation |
|---|---|---|---|---|---|---|
| NODE [8] | | | | | | |
| HNN [26] | ✓ | | | | | |
| LieConv [19] | ✓ | ✓ | | | | |
| DGNet [38] | ✓ | | ✓ | | | ✓ |
| CHNN [20] | ✓ | | | ✓ | | |
| continuous FINDE (proposed) | ✓ | ✓ | ✓ | ✓ | ✓ | |
| discrete FINDE (proposed) | ✓ | ✓ | ✓ | ✓ | ✓ | ✓ |

[38] proposed a model that preserves the total mass of a discretized partial differential equation (PDE). These studies have demonstrated that a neural network with more prior knowledge about first integrals predicts the dynamics of the target system more accurately. See Table 1 for comparison.

Previous studies have mainly attempted to preserve known first integrals. However, in situations where a neural network learns an unknown target system, it is naturally expected that first integrals associated with the target system are also unknown, and it is not clear which of the above methods are available. Given the above, this study proposes *First Integral-preserving Neural Differential Equation* (FINDE) to find and preserve first integrals from data. FINDE has the following advantages.

**Learning First Integrals**  For modeling continuous-time dynamics with known first integrals, many studies have designed architectures or operations of neural networks [13, 19, 20, 26, 38]. For each type of first integral, one dedicated method was proposed. However, the properties of a target system are generally unknown in practice. In contrast, the proposed FINDE finds various kinds of first integrals from data in a unified manner and preserves them in predictions. A symbolic regression confirms that the learned first integrals are consistent with well-known first integrals of target systems.

**Combination with Known First Integrals**  The proposed FINDE can be combined with previously proposed neural networks designed to preserve known first integrals, such as HNN. Therefore, FINDE is available in various situations.

**Exact Preservation of First Integrals**  Even if a first integral is associated with a continuous-time system, it is destroyed after the system is discretized in time for computer simulations. This is true even when using a symplectic integrator, which preserves the system energy only approximately [27]. By leveraging discrete gradients [38], the discrete-time version of FINDE preserves first integrals exactly (up to rounding errors) in discrete time and further improves the prediction performance.

## 2  Background and Related Work

**First Integrals**  Let us consider a time-invariant differential system $\frac{\mathrm{d}}{\mathrm{d}t}\boldsymbol{u} = f(\boldsymbol{u})$ on an $N$-dimensional manifold $\mathcal{M}$, where $\boldsymbol{u}$ denotes the system state and $f : \mathcal{M} \to \mathcal{T}_{\boldsymbol{u}}\mathcal{M}$ represents a vector field on the manifold $\mathcal{M}$. The manifold $\mathcal{M}$ can be $\mathcal{M} = S^1 \times \mathbb{R}^1$ for a pendulum. In this paper, we suppose the manifold $\mathcal{M}$ be a Eucleadian space $\mathbb{R}^N$ for simplicity.

**Definition 1** (first integral). *A quantity $V : \mathcal{M} \to \mathbb{R}$ is referred to as a first integral of a system $\frac{\mathrm{d}}{\mathrm{d}t}\boldsymbol{u} = f(\boldsymbol{u})$ if it remains constant along with any solution $\boldsymbol{u}(t)$, i.e., $\frac{\mathrm{d}}{\mathrm{d}t}V(\boldsymbol{u}) = 0$.*

If a differential system $\frac{\mathrm{d}}{\mathrm{d}t}\boldsymbol{u} = f(\boldsymbol{u})$ is associated with $K$ functionally independent first integrals $V_1, \ldots, V_K$, the solution $\boldsymbol{u}(t)$ given an initial value $\boldsymbol{u}_0$ stays at the $(N-K)$-dimensional submanifold

$$\mathcal{M}' = \{\boldsymbol{u} \in \mathcal{M} : V_1(\boldsymbol{u}) = V_1(\boldsymbol{u}_0), \ldots, V_K(\boldsymbol{u}) = V_K(\boldsymbol{u}_0)\}. \tag{1}$$

The tangent space $\mathcal{T}_{\boldsymbol{u}}\mathcal{M}' \subset \mathcal{T}_{\boldsymbol{u}}\mathcal{M}$ of the submanifold $\mathcal{M}' \subset \mathcal{M}$ at a point $\boldsymbol{u}$ is the orthogonal complement to the space spanned by the gradients $\nabla V_k(\boldsymbol{u})$ of the first integrals $V_k$ for $k = 1, \ldots, K$, that is,

$$\mathcal{T}_{\boldsymbol{u}}\mathcal{M}' = \{\boldsymbol{w} \in \mathcal{T}_{\boldsymbol{u}}\mathcal{M} : \nabla V_k(\boldsymbol{u})^\top \boldsymbol{w} = 0 \text{ for } k = 1, \ldots, K\} \tag{2}$$

If a quantity $V_k$ is a first integral of the system $\frac{d}{dt}\boldsymbol{u} = f(\boldsymbol{u})$, the time-derivative $f$ at point $\boldsymbol{u}$ is on the tangent space $\mathcal{T}_{\boldsymbol{u}}\mathcal{M}'$, being orthogonal to the gradient $\nabla V_k$ of the first integral $V_k$. Then, it holds that $\frac{d}{dt}V_k(\boldsymbol{u}) = \nabla V_k(\boldsymbol{u})^\top \frac{d}{dt}\boldsymbol{u} = \nabla V_k(\boldsymbol{u})^\top f(\boldsymbol{u}) = 0$.

One of the most well-known first integrals is the Hamiltonian $H$, which represents the system energy of a Hamiltonian system. Noether's theorem states that a continuous symmetry of a system leads to a conservation law (and hence a first integral) [27]; a Hamiltonian system is symmetric to translation in time and conserves the Hamiltonian. Symmetries to translation and rotation in space lead to the conservation of linear and angular momenta. Not all first integrals are related to symmetries. A pendulum can be expressed in Cartesian coordinates, and then the rod length constrains the mass position. This kind of constraint is called a holonomic constraint and leads to a first integral. A model for disease spreading called an susceptible-infected-recovered (SIR) model and the dynamics of chemical reactions have the total mass (population) as a first integral. Also for a system described by a PDE, the total mass is sometimes a first integral [23]. See Appendix A for theoretical classification of dynamics.

**First Integrals in Numerical Analysis**   For computer simulations, a differential system is discretized in time and solved by numerical integration. Then, the geometric structures of the system are often destroyed, and most first integrals are no longer preserved. A common remedy is a symplectic integrator, which preserves the symplectic structure and integrates a Hamiltonian system accurately [27]. However, Ge–Marsden theorem states that a symplectic integrator conserves the Hamiltonian only approximately [56]. Hence, many numerical schemes have also been investigated for preserving first integrals exactly, while they cannot preserve the symplectic structure.

Let a superscript $s$ denote the state $\boldsymbol{u}^s$ or time $t^s$ at $s$-th time step, and $\Delta t^s = t^{s+1} - t^s$ denote a time step size. A projection method predicts a next state $\tilde{\boldsymbol{u}}^{s+1}$ from the current state $\boldsymbol{u}^s$ using a numerical integrator and projects it onto the submanifold $\mathcal{M}'$, obtaining the projected state $\boldsymbol{u}^{s+1}$ that preserves the first integrals $V_k$ [24] (see also [27, Section IV.4]). In particular, the projected state $\boldsymbol{u}^{s+1}$ is obtained by solving the optimization problem

$$\boldsymbol{u}^{s+1} = \arg\min_{\boldsymbol{u}'^{s+1}} ||\boldsymbol{u}'^{s+1} - \tilde{\boldsymbol{u}}^{s+1}|| \text{ subject to } V_k(\boldsymbol{u}'^{s+1}) = V_k(\boldsymbol{u}^s) \text{ for } k = 1,\ldots,K. \quad (3)$$

A local coordinate method defines a coordinate system to the neighborhood of the current state $\boldsymbol{u}^s$ and integrates a differential equation on it [43] (see also [27, Section IV.5]). A discrete gradient method defines a discrete analogue to a given differential system and integrates it in discrete time [6, 23, 25, 29, 44, 45]. This method eliminates numerical errors caused by temporal discretization and is used to preserve the Hamiltonian exactly (up to rounding errors) in discrete time.

Except for DGNet, which used discrete gradients to preserve the Hamiltonian [38], all the above methods have never been applied to neural networks due to difficulties that we will introduce later. To our best knowledge, the discrete-time version of FINDE is the first projection method for dynamical systems modeled using neural networks.

**Preservation of First Integrals by Neural Networks**   NODE defines an ODE using a neural network in the most general way with no associated first integrals [8]. NODE is a universal approximator to ODEs [51], and it can approximate any ODE with arbitrary accuracy if there is an infinite amount of training data. In practice, the amount of training data is limited, and prior knowledge about the target system is helpful for learning (see [48] for the case with convolutional neural networks). HNN assumes the target system to be a Hamiltonian system in the canonical form [26]. HNN guarantees various properties of Hamiltonian systems by definition, including the conservation of the energy and the preservation of the symplectic structure in continuous time [27]. Some studies employed a symplectic integrator for HNN to preserve the energy and symplectic structure with smaller numerical errors [10]. LieConv and EMLP-HNN employed neural network architectures with translational and rotational symmetries to preserve momenta [19, 21]. CHNN incorporates a known holonomic constraint in the dynamics [20]. Deep conservation extracts latent dynamics of a PDE system and preserves a quantity of interest by forcing its flux to be zero [34]. HNN++ also guarantees the conservation of the mass in PDE systems by using a coefficient matrix derived from differential operators [38].

Several studies proposed neural networks to learn Lyapunov functions, which are expected to be non-increasing over time, in contrast to first integrals [37, 50]. If the state moves in the direction of

increasing the function, it is projected onto or moved inside the counter line of the gradient of the Lyapunov function. Their idea is similar to the continuous-time version of FINDE but limited to a single non-increasing quantity in continuous time. On the other hand, our proposed FINDE preserves multiple quantities in both continuous and discrete time.

Previous studies aimed to preserve known first integrals. Moreover, except for DGNet [38], all the above methods suffer from numerical errors caused by temporal discretization. In contrast, our proposed FINDE learns first integrals from data and can eliminate discretization errors.

# 3   First Integral-Preserving Neural Differential Equation

The main purpose is to find and preserve first integrals from data by neural networks. We suppose that a target system has at least $K$ unknown functionally independent first integrals. Even when a NODE learns the target system, it is not guaranteed to learn these first integrals. Hence, we introduce a neural network with $K$ outputs, each of which is expected to learn one of first integrals expressed as $V_k : \mathbb{R}^N \to \mathbb{R}$ for $k = 1, \ldots, K$. We denote the set of first integrals by a vector $\boldsymbol{V}(\boldsymbol{u}) = (V_1(\boldsymbol{u}) \; V_2(\boldsymbol{u}) \; \ldots \; V_K(\boldsymbol{u}))^\top$. Then, the submanifold $\mathcal{M}'$ is defined using the neural network $\boldsymbol{V}$ as in Eq. (1).

Because there is no way to define local coordinates on such submanifolds, a local coordinate method is not applicable. When using a projection method, the optimization problem in Eq. (3) should be solved at every training iteration as well as in the prediction phase. Optimization problems are computationally expensive, and common libraries for neural networks do not provide backpropagation algorithms for optimization problems [1, 42].[1] Until a recent study has proposed an algorithm [38], there was no way to obtain discrete gradients of neural networks. Because of these difficulties, no methods for preserving first integrals have been applied to neural networks. By leveraging a projection method and a discrete gradient method, we propose FINDE as follows.

## 3.1   Continuous FINDE: Time-Derivative Projection Method

First, we propose a time-derivative projection method called *continuous FINDE (cFINDE)* for neural networks, which projects the time-derivative onto the tangent space $\mathcal{T}_{\boldsymbol{u}}\mathcal{M}'$. While it still suffers from numerical errors, it is sufficient to find first integrals from data.

We suppose that a neural network called a base model defines the time-derivative $\hat{f} : \mathbb{R}^N \to \mathbb{R}^N$. Then, we define the time-derivative $f$ of the cFINDE $\frac{\mathrm{d}}{\mathrm{d}t}\boldsymbol{u} = f(\boldsymbol{u})$ as

$$f(\boldsymbol{u}) = \hat{f}(\boldsymbol{u}) - \textstyle\sum_{k=1}^K \lambda_k \nabla V_k(\boldsymbol{u}) = \hat{f}(\boldsymbol{u}) - M(\boldsymbol{u})^\top \boldsymbol{\lambda}(\boldsymbol{u}), \tag{4}$$

where $\lambda_k$ is a Lagrange multiplier, $M = \frac{\partial \boldsymbol{V}}{\partial \boldsymbol{u}}$, and $\boldsymbol{\lambda}(\boldsymbol{u}) = (\lambda_1(\boldsymbol{u}) \; \lambda_2(\boldsymbol{u}) \; \ldots \; \lambda_K(\boldsymbol{u}))^\top$. If $V_k$ remains constant,

$$\boldsymbol{0} = \tfrac{\mathrm{d}}{\mathrm{d}t}\boldsymbol{V}(\boldsymbol{u}(t)) = M(\boldsymbol{u})\tfrac{\mathrm{d}}{\mathrm{d}t}\boldsymbol{u} = M(\boldsymbol{u})f(\boldsymbol{u}) = M(\boldsymbol{u})(\hat{f}(\boldsymbol{u}) - M(\boldsymbol{u})^\top \boldsymbol{\lambda}(\boldsymbol{u})), \tag{5}$$

where $\boldsymbol{0} = (0 \; \ldots \; 0)^\top$. By transforming Eq. (5), we obtain the Lagrange multiplier $\boldsymbol{\lambda}(\boldsymbol{u}) = (M(\boldsymbol{u})M(\boldsymbol{u})^\top)^{-1}M(\boldsymbol{u})\hat{f}(\boldsymbol{u})$. By eliminating it, the cFINDE $\frac{\mathrm{d}}{\mathrm{d}t}\boldsymbol{u} = f(\boldsymbol{u})$ is given by

$$f(\boldsymbol{u}) = (I - Y(\boldsymbol{u}))\hat{f}(\boldsymbol{u}) \text{ where } Y(\boldsymbol{u}) = M(\boldsymbol{u})^\top(M(\boldsymbol{u})M(\boldsymbol{u})^\top)^{-1}M(\boldsymbol{u}) \tag{6}$$

**Remark 1** (continuous-time first integral preservation). *The cFINDE $\frac{\mathrm{d}}{\mathrm{d}t}\boldsymbol{u} = f(\boldsymbol{u})$ preserves all first integrals $V_k$ for $k = 1, \ldots, K$ in continuous time, i.e., $\frac{\mathrm{d}}{\mathrm{d}t}V_k = 0$.*

The base model $\hat{f}$ can be a NODE, an HNN, or other models depending on available prior knowledge. Also, if a first integral is already known, one can use it directly as one of first integrals $V_k$ instead of learning it using a neural network. Note that even though the base model $\hat{f}$ is an HNN, due to projection, the cFINDE $f$ is no longer a Hamiltonian system in the strict sense.

Compared to the base model $\hat{f}$, the cFINDE requires the additional computation of the neural network $\boldsymbol{V}$, several matrix multiplications, and an inverse operation. The inverse operation needs a computational cost of $O(K^3)$, which is not costly if the number $K$ of first integrals is small. For satisfying the constraints and geometric structures, many previous models also need the inverse operation, such as Lagrangian neural network (LNN) [13], neural symplectic form [9], and CHNN [20].

---

[1]The algorithm proposed in [2] might work, but it is outside the scope of this paper.

## 3.2 Discrete FINDE: Discrete-Time Projection Method

To eliminate numerical errors caused by temporal discretization, we employ *discrete gradients* and propose a projection method called *discrete FINDE (dFINDE)*.

A discrete gradient $\overline{\nabla}V$ is a discrete analogue to a gradient $\nabla V$ [6, 23, 25, 29, 44, 45]. Recall that a gradient $\nabla V$ of a function $V : \mathbb{R}^N \to \mathbb{R}$ can be regarded as a function $\mathbb{R}^N \to \mathbb{R}^N$ that satisfies the chain rule $\frac{\mathrm{d}}{\mathrm{d}t}V(\boldsymbol{u}) = \nabla V(\boldsymbol{u})^\top \frac{\mathrm{d}}{\mathrm{d}t}\boldsymbol{u}$. Analogously, a discrete gradient $\overline{\nabla}$ is defined as follows.

**Definition 2** (discrete gradient). *A discrete gradient $\overline{\nabla}V$ of a function $V : \mathbb{R}^N \to \mathbb{R}$ is a function $\mathbb{R}^N \times \mathbb{R}^N \to \mathbb{R}^N$ that satisfies*

$$V(\boldsymbol{v}) - V(\boldsymbol{u}) = \overline{\nabla}V(\boldsymbol{v}, \boldsymbol{u})^\top(\boldsymbol{v} - \boldsymbol{u}) \ \text{ and } \ \overline{\nabla}V(\boldsymbol{u}, \boldsymbol{u}) = \nabla V(\boldsymbol{u}). \tag{7}$$

The first condition is a discrete analogue to the chain rule when replacing the time-derivatives $\frac{\mathrm{d}}{\mathrm{d}t}V$ and $\frac{\mathrm{d}}{\mathrm{d}t}\boldsymbol{u}$ with finite differences $(V(\boldsymbol{v}) - V(\boldsymbol{u}))$ and $(\boldsymbol{v} - \boldsymbol{u})$, respectively, and the second condition ensures the consistency with the ordinary gradient $\nabla V$. A discrete gradient $\overline{\nabla}V$ is not uniquely determined and has been obtained manually. Recently, the automatic discrete differentiation algorithm (ADDA) has been proposed in [38], which obtains a discrete gradient of a neural network in a similar way to the automatic differentiation algorithm [1, 42]. The discrete gradient is defined in discrete time, and hence a numerical integration using the discrete gradient is free from numerical errors caused by temporal discretization. See Appendix B and the references [6, 23, 38] for more details.

Following [11, 15], we introduce a discrete analogue to the tangent space $\mathcal{T}_{\boldsymbol{u}}\mathcal{M}'$ called the discrete tangent space $\mathcal{T}_{(\boldsymbol{v},\boldsymbol{u})}\mathcal{M}'$. In particular, for a pair $(\boldsymbol{v}, \boldsymbol{u}) \in \mathcal{M}'$ of points, it is defined as

$$\mathcal{T}_{(\boldsymbol{v},\boldsymbol{u})}\mathcal{M}' = \{\boldsymbol{w} \in \mathbb{R}^N : \overline{\nabla}V_k(\boldsymbol{v}, \boldsymbol{u})^\top \boldsymbol{w} = 0 \text{ for } k = 1, \dots, K\}. \tag{8}$$

If the finite difference $(\boldsymbol{u}^{s+1} - \boldsymbol{u}^s)$ between the predicted and current states is on the discrete tangent space $\mathcal{T}_{(\boldsymbol{u}^{s+1},\boldsymbol{u}^s)}\mathcal{M}'$, the first integrals $V_k$ are preserved because $V_k(\boldsymbol{u}^{s+1}) - V_k(\boldsymbol{u}^s) = \overline{\nabla}V_k(\boldsymbol{u}^{s+1}, \boldsymbol{u}^s)^\top(\boldsymbol{u}^{s+1} - \boldsymbol{u}^s) = 0$. Note that similar concepts defined in different ways are also referred to as discrete tangent spaces [14, 16].

Let $\hat{\psi}$ denote a discrete-time base model that satisfies $\frac{\tilde{\boldsymbol{u}}^{s+1} - \boldsymbol{u}^s}{\Delta t^s} = \hat{\psi}(\boldsymbol{u}^s; \Delta t^s)$, where $\tilde{\boldsymbol{u}}^{s+1}$ denotes the predicted state. We assume that the base model $\hat{\psi}$ is composed of a continuous-time base model $\hat{f}$ and a numerical integrator. Then, the dFINDE $\frac{\boldsymbol{u}^{s+1} - \boldsymbol{u}^s}{\Delta t^s} = \psi(\boldsymbol{u}^{s+1}, \boldsymbol{u}^s; \Delta t^s)$ is given by

$$\psi(\boldsymbol{u}^{s+1}, \boldsymbol{u}^s; \Delta t^s) = \hat{\psi}(\boldsymbol{u}^s; \Delta t^s) - \overline{M}(\boldsymbol{u}^{s+1}, \boldsymbol{u}^s)^\top \boldsymbol{\lambda}(\boldsymbol{u}^{s+1}, \boldsymbol{u}^s), \tag{9}$$

where $\overline{M}(\boldsymbol{u}^{s+1}, \boldsymbol{u}^s) = (\overline{\nabla}V_1(\boldsymbol{u}^{s+1}, \boldsymbol{u}^s) \ \dots \ \overline{\nabla}V_K(\boldsymbol{u}^{s+1}, \boldsymbol{u}^s))^\top$. As is the case in continuous time, the preservation of the first integrals $V_k$ leads to

$$\boldsymbol{0} = \frac{\boldsymbol{V}(\boldsymbol{u}^{s+1}) - \boldsymbol{V}(\boldsymbol{u}^s)}{\Delta t^s} = \overline{M}(\boldsymbol{u}^{s+1}, \boldsymbol{u}^s)\frac{\boldsymbol{u}^{s+1} - \boldsymbol{u}^s}{\Delta t^s} = \overline{M}(\boldsymbol{u}^{s+1}, \boldsymbol{u}^s)\psi(\boldsymbol{u}^{s+1}, \boldsymbol{u}^s; \Delta t^s). \tag{10}$$

Substituting Eq. (9) and eliminating the Lagrange multiplier $\boldsymbol{\lambda}$, we obtain

$$\psi(\boldsymbol{u}^{s+1}, \boldsymbol{u}^s; \Delta t^s) = (I - \overline{Y}(\boldsymbol{u}^{s+1}, \boldsymbol{u}^s))\hat{\psi}(\boldsymbol{u}^s; \Delta t^s) \text{ where } \overline{Y} = \overline{M}^\top(\overline{M}\ \overline{M}^\top)^{-1}\overline{M}. \tag{11}$$

**Remark 2** (discrete-time first integral preservation). *The dFINDE $\frac{\boldsymbol{u}^{s+1} - \boldsymbol{u}^s}{\Delta t^s} = \psi(\boldsymbol{u}^{s+1}, \boldsymbol{u}^s; \Delta t^s)$ preserves all first integrals $V_k$ for $k = 1, \dots, K$ in discrete time, i.e., $V_k(\boldsymbol{u}^{s+1}) - V_k(\boldsymbol{u}^s) = 0$.*

Due to projection, dFINDE can be regarded as a projection method using discrete gradients. For the base model $\hat{\psi}$, the continuous-time base model $\hat{f}$ can be a NODE, an HNN, or other models, and the numerical integrator can be a Runge–Kutta method, the leapfrog integrator, or others.

Because dFINDE is an implicit method, it is computationally expensive for prediction. However, the next state $\boldsymbol{u}^{s+1}$ is given for training, and the ADDA can explicitly obtain the discrete gradient w.r.t. the pair $(\boldsymbol{u}^{s+1}, \boldsymbol{u}^s)$ as well as its computational graph. Thus, dFINDE can be computed explicitly and optimized by standard backpropagation algorithms. Moreover, we suppose that dFINDE projects the finite difference $\hat{\psi}$ only at every time step, whereas cFINDE projects the time-derivative $\hat{f}$ at every substep inside a numerical integrator. Therefore, dFINDE is less computationally expensive than cFINDE for training. In contrast, a typical projection method requires much computational cost to solve an optimization problem for training, and standard backpropagation algorithms are not applicable to it.

Table 2: Datasets, Dynamics, and First Integrals.

| | | | First Integrals | | | |
|---|---|---|---|---|---|---|
| Dataset | Dynamics | $N$ | Energy | Momentum | Mass | Constraint |
| Two-body problem | Canonical Hamiltonian | 8 | ✓ | ✓ | | |
| Discretized KdV equation | Non-canonical Hamiltonian | 50 | ✓ | | ✓ | |
| Double pendulum | Poisson | 8 | ✓ | | | ✓ |
| FitzHugh–Nagumo model | Dirac | 4 | | | | ✓ |

**Remark 3** (trainability). *The dFINDE can be trained using the standard backpropagation algorithm, whereas a straightforward application of a projection method cannot.*

## 4 Experiments

### 4.1 Experimental Settings

**Target Systems**  We evaluated FINDE and base models using datasets associated with first integrals, summarized in Table 2. A gravitational two-body problem (2-body) on a 2-dimensional configuration space is a typical Hamiltonian system in the canonical form. In addition to the total energy, it has first integrals related to symmetries in space, namely, the linear and angular momenta. The Korteweg–De Vries (KdV) equation is a PDE model of shallow water waves. This is a Hamiltonian system in a non-canonical form and has the Hamiltonian, total mass, and many other quantities as first integrals. A double pendulum (2-pend) is a Hamiltonian system in polar coordinates. However, we transformed it to Cartesian coordinates; it was no longer a Hamiltonian system but a Poisson system. The lengths of two rods work as holonomic constraints and lead to four first integrals. The FitzHugh–Nagumo model is a biological neuron model as an electric circuit, which exhibits a rapid and transient change of voltage called a spike. As an electric circuit, the currents through and voltages applied to the inductor and capacitor can be regarded as system states, and the states are constrained by the circuit topology and Kirchhoff's current and voltage laws. Then, this system has a state of four elements and two first integrals. Due to energy dissipation in the resistor, the model is not a Poisson system, but one can find a Dirac structure [53]. See Appendix C for more details.

**Implementation**  We implemented the proposed FINDE and evaluated it under the following settings. We implemented all codes by modifying the officially released codes of HNN [26] [2] and DGNet [38][3]. We used Python v3.8.12 with packages scipy v1.7.3, pytorch v1.10.2, torchdiffeq v0.1.1, functorch v1.10 preview, and gplearn v0.4.2. We used the Dormand–Prince method (dopri5) [18] as the numerical integrator, unless otherwise stated. All experiments were performed on a single NVIDIA A100 provided by (ANONYMOUS PROVIDER).

Following HNN [26] and DGNet [38], we represented the first integrals $V$, NODE, and HNN $H$ using fully-connected neural networks with two hidden layers. Each hidden layer had 200 units and preceded a hyperbolic tangent activation function. Each weight matrix was initialized as an orthogonal matrix. The input was the state $u$, and the output represented the first integrals $V$ for FINDE, time-derivative $\hat{f}$ for NODE, and the Hamiltonian $H$ for HNN. For the KdV dataset, we used a 1-dimensional convolutional neural network (CNN), each of whose layers had a kernel size of 3. The double pendulum is a second–order system, implying that the time-derivative $\frac{\mathrm{d}}{\mathrm{d}t}q$ of the position $q$ is known as the velocity $v$. Hence, we treated only the acceleration $\frac{\mathrm{d}}{\mathrm{d}t}v$ as the output to learn. This assumption slightly improved the absolute performances but did not change the relative trends.

We used the *1-step* error as the loss function to be minimized. In particular, it is the mean squared error (MSE) between the ground truth state $u_{\mathrm{GT}}^s$ and the state $u_{\mathrm{pred.}}^s$ predicted from the previous step $u_{\mathrm{GT}}^{s-1}$. The base model and FINDE were jointly trained using the Adam optimizer [33] with the parameters $(\beta_1, \beta_2) = (0.9, 0.999)$ and a batch size of 200. The learning rate was initialized to $10^{-3}$ and decayed to zero with a cosine annealing [36].

---

[2]`https://github.com/greydanus/hamiltonian-nn` (Apache-2.0 License)

[3]`https://github.com/tksmatsubara/discrete-autograd` (MIT License)

**Evaluation Metric** As an evaluation metric, we used the 1-step error, which is identical to the loss function. We displayed it at the scale of $\times 10^{-9}$. The lower this indicator, the better, as emphasized by $\downarrow$. While several studies used the MSEs of the state or system energy over the whole time series [26, 38], we consider these indicators are misleading, as pointed in several studies [4]. For example, in the case of a periodic orbit, an orbit that is correctly learned except for a slight difference in angular velocity will have the same MSE as an orbit that never moves from its initial position. Instead, we used the valid prediction time (*VPT*) [4, 32, 54]. VPT denotes the time point $s$ divided by the length $S$ of time series at which the MSE of the predicted state $\boldsymbol{u}^s_{\text{pred.}}$ exceeds a given threshold $\theta$ for the first time in an initial value problem, that is,

$$VPT(\boldsymbol{u}_{\text{pred.}}; \boldsymbol{u}_{\text{GT}}) = \frac{1}{S} \arg\max_{s_f} \{s_f | \text{MSE}(\boldsymbol{u}^s_{\text{pred.}}, \boldsymbol{u}^s_{\text{GT}}) < \theta \text{ for all } s \leq s_f\}. \quad (12)$$

To obtain VPTs, we normalized each element of state to have zero mean and unit variance in the training data and set $\theta$ to 0.01. The higher this indicator, the better, as emphasized by $\uparrow$. Because of the "spiking" behavior of the FitzHugh–Nagumo model, a small error in phase is regarded as a large error in state. To measure the qualitative performance, we calculated VPTs by allowing for a delay and advance of up to 5 steps.

## 4.2 First Integral Preservation for Hamiltonian System

Before learning first integrals from data, we first evaluated FINDE as a numerical integrator using a known mass-spring system. The system has the state $\boldsymbol{u} = (q\ v)^\top$, the dynamics $\frac{\mathrm{d}}{\mathrm{d}t}q = v$ and $\frac{\mathrm{d}}{\mathrm{d}t}v = -q$, and the system energy $E(q,v) = \frac{1}{2}(q^2 + v^2)$. Using the initial value $(1.0\ 0.0)^\top$ and the time step size $\Delta t = 0.2$, we solved the initial value problem of the true ODE using the leapfrog integrator. We applied FINDE with the true system energy $E$ as the first integral $V$. Note that no neural networks nor training were involved.

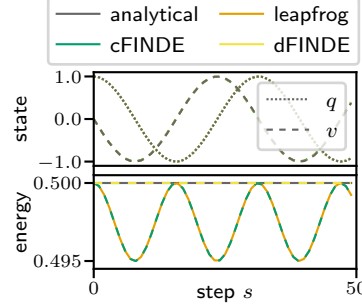

Figure 1: Integration of a known mass-spring system.

The results with the analytical solution are shown in Fig. 1. The upper panel shows that the time series predicted by comparison methods overlap each other and are apparently almost identical. However, the lower panel shows that the energy obtained from the states predicted by the leapfrog integrator is fluctuating. The same is true for the case with the cFINDE. This is because the symplectic integrator and the cFINDE suffer from numerical errors caused by temporal discretization. In contrast, the dFINDE preserves the energy accurately. This is because, at every step, the dFINDE projects the state $(q\ v)^\top$ onto the discrete tangent space $\mathcal{T}_{(\boldsymbol{v},\boldsymbol{u})}\mathcal{M}'$. Although a smaller step size reduces numerical errors, this result demonstrates the advantage of dFINDE.

## 4.3 Learning First Integrals from Data of Hamiltonian System

We evaluated FINDE on learning from the 2-body dataset. We used HNN as the base model $\hat{f}$. We found that the FINDE got better performances if it did not treat the Hamiltonian $H$ of the HNN as one of first integrals $V_k$. The medians and standard deviations of 5 trials are summarized in the leftmost column of Table 3. The cFINDE achieved better VPTs than the vanilla HNN with $K = 1$ to 2, and the performance was suddenly degraded for $K = 3$. The dFINDE showed a similar trend with slightly better performances. The HNN with FINDE found two first integrals in addition to the Hamiltonian $H$ of the HNN. Even though a two-body problem is a Hamiltonian system that HNN can learn, the prior knowledge that there exist first integrals other than the Hamiltonian $H$ can be a clue to better learning. The HNN with FINDE got worse 1-step errors, suggesting that without FINDE, HNN overfitted short-term change and had difficulty predicting long-term dynamics.

We performed a symbolic regression of first integrals $V$ learned by the neural network. For $K = 2$, the learned first integrals $V$ were identical to the linear momenta in the $x$- and $y$-directions up to affine transformation in most cases. See Appendix D.1 for more details.

We depict example results in Fig. 2. In the absence of FINDE, the mass positions $(x_1, y_1), (x_2, y_2)$ became inaccurate in a short time and the center-of-gravity position $(x_c, y_c) = (\frac{x_1+x_2}{2}, \frac{y_1+y_2}{2})$ deviated rapidly. The HNN with cFINDE accurately predicted the state for a longer period. Even after errors in the mass positions became non-negligible, errors in the center-of-gravity position were still small. We show the absolute errors averaged over all trials in Fig. 3. In each of $x$- and $y$-directions,

Table 3: Results of FINDE.

| Model | $K$ | 2-body + HNN | | KdV | | 2-pend | | FitzHugh–Nagumo | |
|---|---|---|---|---|---|---|---|---|---|
| | | 1-step↓ | VPT↑ | 1-step↓ | VPT↑ | 1-step↓ | VPT↑ | 1-step↓ | VPT↑ |
| base model | – | 5.17 ±0.57 | 0.362 ±0.026 | 5.59 ±0.30 | 0.339 ±0.038 | 0.82 ±0.02 | 0.110 ±0.035 | 73.66 ±12.59 | 0.236 ±0.053 |
| + cFINDE | 1 | 7.10 ±1.25 | 0.374 ±0.036 | 6.24 ±0.44 | 0.371 ±0.088 | 0.75 ±0.04 | 0.156 ±0.042 | 54.18 ±8.12 | 0.127 ±0.148 |
| | 2 | 7.78 ±1.39 | **0.450** ±0.052 | **2.59** ±0.11 | 0.608 ±0.085 | 0.73 ±0.05 | 0.198 ±0.088 | **37.03** ±3.81 | **0.437** ±0.084 |
| | 3 | $>10^3$ | 0.147 ±0.146* | 3.19 ±0.37 | **0.730** ±0.091 | **0.69** ±0.03 | 0.411 ±0.093 | $>10^6$ | 0.007 ±0.007* |
| | 4 | $>10^3$ | 0.101 ±0.005 | 3.65 ±0.30 | 0.641 ±0.071 | 0.77 ±0.07 | 0.395 ±0.083 | — | |
| | 5 | $>10^3$ | 0.080 ±0.014 | 4.68 ±0.43 | 0.601 ±0.069 | 0.80 ±0.07 | **0.585** ±0.097 | — | |
| | 6 | $>10^3$ | 0.070 ±0.019 | 7.79 ±0.51 | 0.425 ±0.067 | 12.53 ±0.00 | 0.005 ±0.000* | — | |
| + dFINDE | 1 | 7.01 ±1.06 | 0.379 ±0.040 | 11.61 ±6.60 | 0.288 ±0.083 | 0.75 ±0.10 | 0.152 ±0.017 | 47.07 ±8.03 | 0.117 ±0.122 |
| | 2 | 7.03 ±1.00 | **0.475** ±0.022 | **2.70** ±0.26 | 0.598 ±0.059 | 0.74 ±0.05 | 0.271 ±0.111 | **33.24** ±3.40 | **0.455** ±0.032 |
| | 3 | 54.78 ±36.39 | 0.309 ±0.024 | 3.78 ±0.27 | 0.636 ±0.024 | **0.69** ±0.05 | 0.447 ±0.081 | 319.70 ±91.11 | 0.049 ±0.007 |
| | 4 | $>10^3$ | 0.102 ±0.015 | 3.48 ±0.32 | **0.780** ±0.059 | 0.71 ±0.03 | 0.454 ±0.060 | — | |
| | 5 | $>10^3$ | 0.086 ±0.011* | 5.26 ±0.15 | 0.718 ±0.038 | 0.86 ±0.09 | **0.591** ±0.087 | — | |
| | 6 | $>10^3$ | 0.059 ±0.017 | 9.60 ±3.61 | 0.573 ±0.121 | 58.88 ±22.98 | 0.037 ±0.039 | — | |

A standard deviation follows ± symbol. Underlines indicate results better than the base models' results, and bolded fonts indicate the best results. * denotes that some trials failed in training because of the underflow of the step size. A dash denotes a case we did not try.

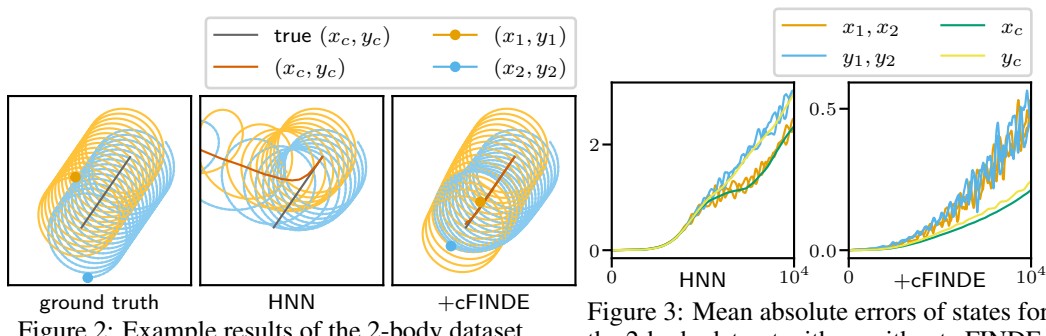

Figure 2: Example results of the 2-body dataset.

Figure 3: Mean absolute errors of states for the 2-body dataset with or without cFINDE.

the HNN without FINDE produced errors in the center-of-gravity position $x_c$ (or $y_c$) and those in the mass positions $x_1, x_2$ (or $y_1, y_2$) at almost the same level. In contrast, when the cFINDE is present, errors in the center-of-gravity position were much smaller than those in the mass positions, implying that errors in one mass position canceled out errors in the other mass position.

Therefore, we conclude that FINDE not only had better prediction accuracy but also found and preserved linear momenta (which are related to symmetries in space) more accurately despite not having prior knowledge about symmetries.

## 4.4 Learning First Integrals from Data of Unknown Systems

It is often unclear whether a target system is a Hamiltonian system or not, but one can expect that the target system has several first integrals. We evaluated FINDE using NODE as the base model. We summarized the results in Table 3.

For the KdV dataset, the NODE with FINDE got much better 1-step errors and VPTs for a wide range of $K$. Figure 4 shows an example result. The top panels show that the prediction results were apparently similar. The bottom panels summarize mean absolute errors in states $\boldsymbol{u}$, total mass $\sum_k u_k$, and energy. In the absence of FINDE, the NODE increased all of its errors in proportion to time. With the cFINDE, the error in total mass increased at the point where the two solitons collided but then returned to the original level. Although the calculation is slightly inaccurate, the cFINDE learned to preserve the total mass. The rightmost panel shows that the error in energy continued to increase for $K = 2$, but it stayed within a small range for $K = 3$. These results suggest that the first or second quantity learned by the cFINDE was total mass, the third quantity was system energy, and the remaining quantity may correspond to one of the many first integrals of the KdV equation.

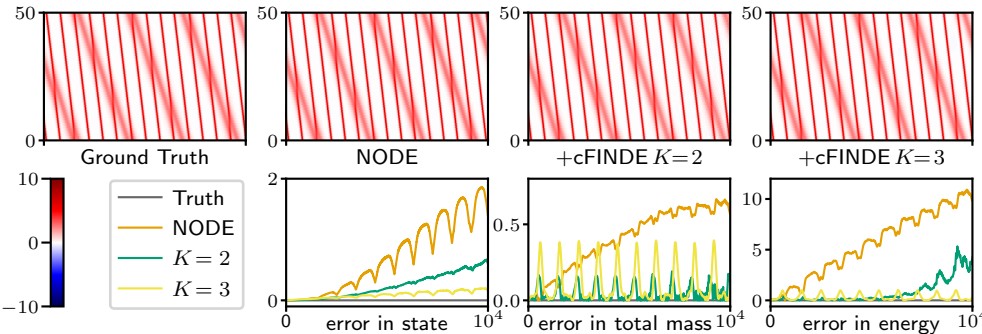

Figure 4: Example results of the KdV dataset. (top) Predicted states. Red belts denote moving solitons. (bottom) Mean absolute errors.

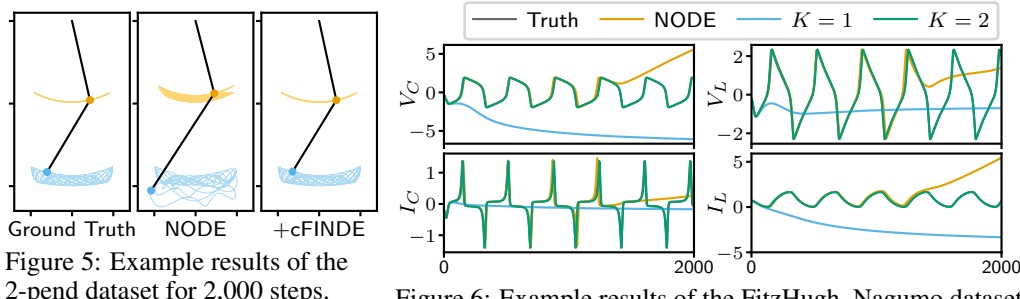

Figure 5: Example results of the 2-pend dataset for 2,000 steps.

Figure 6: Example results of the FitzHugh–Nagumo dataset.

For the 2-pend dataset, the NODE with FINDE got better 1-step errors and VPTs for $K = 1$ to 5 except for the 1-step error of the dFINDE with $K = 5$. In addition to the system energy, the double pendulum has two holonomic constraints on the position, which lead to two additional constraints involving the velocity (see Appendix C for details). Thus, it is reasonable that the NODE with FINDE got the best VPTs for $K = 5$ first integrals and totally failed when assuming $K > 5$ first integrals. As exemplified in Fig. 5, the NODE without FINDE did not preserve the lengths of rods, making the states deviate gradually. See Appendix D.2 for the case when actual constraints are known. For the FitzHugh–Nagumo dataset, the NODE with FINDE got much better 1-step errors and VPTs for $K = 2$. As exemplified in Fig. 6, the ground truth state converged to a periodic orbit, and only the NODE with cFINDE for $K = 2$ reproduced such dynamics. On the other hand, the state did not stay at a limited region without FINDE and converged to a wrong equilibrium with the cFINDE for $K = 1$. For $K = 1$, the sole quantity $V_1$ may have tried to learn both of the two first integrals and remained under-trained. In these two cases, FINDE found all first integrals; $K = 5$ for the 2-pend dataset and $K = 2$ for the FitzHugh–Nagumo dataset.

## 5 Conclusion

This study proposed *first integral-preserving neural differential equation* (FINDE). FINDE projects the time evolution onto the submanifold defined using the (discrete) gradients of first integrals represented by a neural network. With an appropriate number of assumed first integrals, FINDE predicted future states more accurately than base models. Not only that, FINDE found and preserved the system energy and the total mass as first integrals, first integrals related to symmetries in space, and first integrals led by constraints in a unified manner. Therefore, FINDE has the potential to make a scientific discovery by revealing unknown properties of target dynamical systems.

The 1-step errors were on the order of $10^{-5}$ to $10^{-4}$ in absolute error, being much larger than the numerical error tolerance of $10^{-9}$ used in the experiments; numerical errors were negligible compared to modeling errors. However, the dFINDE tended to get VPTs better than the cFINDE despite the fact that its advantage is to eliminate numerical errors caused by temporal discretization. This result suggests that a method leading to smaller numerical errors results in a model with smaller modeling errors. Similar tendencies have been observed in previous works [10, 38], and these results may form a new frontier for integrating numerical and modeling errors.

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
