# OpenReview forum: "FINDE: Neural Differential Equations for Finding and Preserving Invariant Quantities"
_NeurIPS.cc/2022/Conference — NeurIPS 2022 Submitted_

### Official Review · Reviewer_fysu · 2022-06-23

**Rating:** 7
**Confidence:** 4
**Soundness:** 3 good
**Presentation:** 3 good
**Contribution:** 3 good

**Summary:**

The objective of the paper is to propose a method that enables to find (and preserve) invariant quantities in the observations of a dynamical system.

The authors proposed in their work  the joint learning of a derivative ($f$) approximating the dynamics of the observations and parameterized first integrals $V=(V_1,..., V_k)$  that are used between each prediction step. Several technical developments are conducted to enforce the respect of the first integrals.


### POST REBUTTAL ###

I would like to thanks the authors for their rebuttal statements. The proposed explanations details and submission modification for the unclear parts are satisfying (notably on the projection and the discretization errors).  Therefore, I maintain my score.


**Questions:**

1) Section 3.1 provides the dynamics that respects the invariants given $\hat{f}$ and lagrange multipliers. I suggest for clarity to present the optimization problem associated with these $\lambda$.

2) Can the authors also detail why the continuous version is subject to numerical errors compared to the discrete ones ?

3) What is the method used to solve eq.3 (at each iteration) ? Or is the integration performed using the “modified” (i.e c/d Finde) version of the learned dynamics ?

4) L.144: it is unclear to me why gradients are not computable. Indeed solving eq.3 can be thought of as a meta-learning problem (yet the optimization would not be exact).

5) I get that the whole point of the method is to provide a way to “respect” first integrals, i.e. to “alter” an initial integrator so that $(V_1,..., V_k)$  are constant. However, since it is done pointwise it allows potential spikes in the learned first integral as for instance illustrated in fig.4. Can the authors comment on this and potential implications (for instance it severely arms the interpretability of the learned first integral )? Can the authors think of any additional constraints or inductive biases to palliate such drawback ?

6) What is the influence of the hyper-parameters on the learning mainly on the relative influence of the LR of the networks associated with the first integral ?


**Limitations:**

see questions.

**Strengths And Weaknesses:**

S:  The paper and its objective are clearly presented and very relevant to the community and the development of further methods.
Moreover,  the paper is well-written and the technical contributions of the paper are clear.


W: I identify two major weaknesses.
The first one lies in the presentation:  the loss objective is not clearly written in the main text which slightly hampers the understanding of the text. Indeed, the authors remain quite far from the application to focus on their technical contribution.Far from being essential, I nonetheless highly recommend the inclusion of an “algorithm” section to detail how the first integrals are learned practically using for instance NN.

The experiments are conducted using low dimensional data (and it presents a clear case of the potential of the proposition in this case) but the application of the proposition to  higher dimension data is not answered.

---

> ### Author Response · Authors · 2022-08-01
> **Here is our responses to the weaknesses.**
>
> **Weakness 1-1**
> > I identify two major weaknesses. The first one lies in the presentation: the loss objective is not clearly written in the main text which slightly hampers the understanding of the text.
>
> Sorry for our inadequate explanation.
> We have added Appendix B.3 to the revised supplementary PDF file and written the details of loss functions as follows.
>
> Loss Function:
> For the cFINDE, a state $u_{GT}^{s}$ is taken from the training dataset, and a numerical integrator solves the cFINDE $\dot u=f(u)$ and predicts the next state $u_{pred}^{s+1}$.
> Then, the cFINDE is trained to minimize the difference between the predicted state $u_{pred}^{s+1}$ and the ground truth $u_{GT}^{s+1}$ taken from the training dataset.
> Instead of using the state directly, we used the finite difference normalized by the time step size $\Delta t^s$ for the loss function;
> $$
> ||\frac{u_{GT}^{s+1}-u_{GT}^s}{\Delta t^s}-\frac{u_{pred}^{s+1}-u_{GT}^s}{\Delta t^s}||^2_2.
> $$
>
> For the dFINDE, we have defined the state update in Eq. (11).
> Given a current state $u^{s}$, the process to obtain the next state $u^{s+1}$ is implicit.
> Therefore, the prediction by the dFINDE is implicit.
> However, during the training phase, the ground truth $u_{GT}^{s+1}$ of the next state is known.
> Hence, we assigned the data points $u_{GT}^{s}$ and $u_{GT}^{s+1}$ in the training dataset to both the current state $u^{s}$ and the next state $u^{s+1}$.
> Then, the loss function can be the difference between the left- and right-hand sides, that is,
> $$
> ||\frac{u_{GT}^{s+1}-u_{GT}^{s}}{\Delta t^s}-(I-\overline{Y}(u_{GT}^{s+1},u_{GT}^{s}))\hat\psi(u_{GT}^{s};\Delta t^s)||^2_2.
> $$
> The discrete Jacobian $\overline{M}$ (and hence $\overline{Y}$) can be obtained explicitly, and an explicit solver can be used as the numerical integration $\hat\psi$.
> Hence, the process to get the value of the loss function is explicit, and the dFINDE can be trained in an explicit way, whereas the prediction is still in an implicit way.
>
> **Weakness 1-2**
> > Indeed, the authors remain quite far from the application to focus on their technical contribution. Far from being essential, I nonetheless highly recommend the inclusion of an “algorithm” section to detail how the first integrals are learned practically using for instance NN.
>
> Thank you for your suggestion.
> In the final version, we will create Algorithm section and introduce the algorithm in detail.
>
> **Weakness 2**
> > The experiments are conducted using low dimensional data (and it presents a clear case of the potential of the proposition in this case) but the application of the proposition to higher dimension data is not answered.
>
> Sorry for our unclear presentation, but we have already conducted experiments with the KdV dataset that has a state of 50 dimensions (Table 2).
> Even with 50-dimensional data, the introduction of only two to three first integrals significantly improves the prediction performance (Table 3).
> Our proposed FINDE is valid for higher dimension data.
>
> We will clearly state the above fact in the final version.

---

> ### Author Response · Authors · 2022-08-01
> **Thank you for insightful comments. Here is our responses to your questions.**
>
> We would like to thank you for your insightful comments. We are pleased that you considered our method clearly presented and very relevant to the community. We would like to address all your concerns below.
>
> **Question 1**
> > 1. Section 3.1 provides the dynamics that respects the invariants given and lagrange multipliers. I suggest for clarity to present the optimization problem associated with these $\lambda$.
>
> Thank you for your suggestion.
> A projection method is defined as an optimization method, and the Lagrangian multiplier $\lambda$ appears therein.
> The cFINDE can be defined by taking the limit of a projection method as the time interval approaches zero.
>
> We have added Appendix B.1 to the supplementary PDF file to introduce the derivation of FINDE and the related optimization problem.
>
> **Question 2**
> > 2. Can the authors also detail why the continuous version is subject to numerical errors compared to the discrete ones ?
>
> The cFINDE is still an ODE and requires a numerical integrator to predict future states.
> This fact implies that the cFINDE suffers from numerical errors caused by temporal discretization.
> In contrast, the dFINDE is defined using a finite difference, implying that the dFINDE is inherently a discrete-time model and can predict future states without further discretization.
> In this sense, the discrete version is not subject to numerical (temporal discretization) errors.
>
> To clear up confusion, we will add the above explanations to Section 3.
>
>
> **Question 3**
> > 3. What is the method used to solve eq.3 (at each iteration) ? Or is the integration performed using the “modified” (i.e c/d Finde) version of the learned dynamics ?
>
> Eq. (3) is a general form of projection method, and the method of Lagrange multipliers is often used to solve it.
> We have provided the derivation in the revised supplementary PDF file.
> Since the purpose of projection and FINDE are the same, it is meaningless to use them in combination.
> The neural projection method [62] used a gradient descent method by assuming the constraint $V(x)=0$.
> This assumption holds for specific cases (e.g., holonomic constraints in a fixed environment), but not for most first integrals $V(u)=const$, whose values vary across initial conditions.
>
> **Question 4**
> > 4. L.144: it is unclear to me why gradients are not computable. Indeed solving eq.3 can be thought of as a meta-learning problem (yet the optimization would not be exact).
>
> At L.144, we mentioned not Eq. (3) but a discrete gradient method, and what we meant is that the reference [38] is the first study to provide an algorithm to obtain the "discrete" gradient of a neural network.
> We will rephrase the sentences to clarify our intent.
>
> **Question 5**
> > 5. I get that the whole point of the method is to provide a way to “respect” first integrals, i.e. to “alter” an initial integrator so that $(V_1,\dots,V_k)$ are constant. However, since it is done pointwise it allows potential spikes in the learned first integral as for instance illustrated in fig.4. Can the authors comment on this and potential implications (for instance it severely arms the interpretability of the learned first integral )? Can the authors think of any additional constraints or inductive biases to palliate such drawback ?
>
> We appreciate your correct understanding and insightful comment.
> Our method is done pointwise, and the learned first integral is not always a simple function that is consistent over the domain.
> The same can be said about the energy function of HNN, and this type of problem is an open problem for neural network models of dynamical systems.
>
> Figure 2 shows that the learned first integral does not perfectly match the momentum, but Table A1 shows that symbolic regression yielded equations that perfectly match the momentum in most cases.
> As long as the truth equation is of lower-order, symbolic regression ignores higher-order nuisance terms and potentially finds the true equation.
>
> **Question 6**
> > 6. What is the influence of the hyper-parameters on the learning mainly on the relative influence of the LR of the networks associated with the first integral ?
>
> We roughly explored the hyperparameters, namely the learning rate, the number of iterations, and whether to use the cosine annealing, and found that the proposed FINDE was always superior to the base model.
> It is an interesting idea to separately adjust the learning rate $V$ of the network for first integrals.

---

### Official Review · Reviewer_kyoV · 2022-06-27

**Rating:** 6
**Confidence:** 5
**Soundness:** 3 good
**Presentation:** 3 good
**Contribution:** 3 good

**Summary:**

The authors propose modify the Neural ODE model to conserve the first integrals of the physical system, where the  first integrals are either known or learned by a neural network.

**Questions:**

- Can the authors clarify the difference between the cFINDE and dFINDE? cFINDE is derived for continuous functions f(u) and V(u), but in practice, it surely uses an integrator to compute the ODE solution. dFINDE also uses a numerical integrator for Psi: “We assume that the base model ψ is composed of a continuous-time base model and a numerical integrator.” Is the only difference between the models in the computation of the grad_V (u)?

- In case first integrals are learned, how do you prevent it from learning degenerate solutions, e.g. predicting zero function for all inputs u? My understanding is that function f(u) is trained together with the learned first integrals V(u). In the best case, the function f(u) can already preserve all or some of the first integrals, and V(u) can only learn a degenerate function.

- In Figure 1 authors demonstrate that leapfrog integrator does not preserve the total energy, while dFINDE does, while dFINDE uses Dormand–Prince method (dopri5)  (mentioned in section 4.1). Presumably, the ability to conserve the integral of dFINDE heavily depends on the integrator used. It is unfair to compare dFINDE equipped with dopri5 versus leapfrog integrator. Can authors provide the results or hypothesise about the following comparisons: 1) leapfrog versus dFINDE with leapfrog integrator 2) dopri5 versus on dFINDE with dopri5 integrator?

- Have authors tested the model on the systems with different masses and total energy than on the training set to verify that the learned constraints indeed learn to compute the  total energy/mass that need to be conserved?


**Limitations:**

I strongly suggest the authors to acknowledge the limitations of their work.

Some limitations worth mentioning in the paper:
- The ability of the cFINDE and dFINDE to preserve the first integrals depends on the numerical integrators. dFINDE also depends on the time discretisation delta_t.
- The results in Table 3 suggest that the model is sensitive to K — the number of the first integrals chosen. Moreover, with K>3 the predictions “explode” in case of 2-body in
- FitzHugh–Nagumo systems.
- By construction, enforces that certain quantities of the system have to be conserved exactly. The model is not applicable for the systems that are conserving, e.g. having dissipative energy.


**Strengths And Weaknesses:**

The authors propose a neat way to introduce the conservation constraint by adding the Lagrange multipliers to the derivative function of the Neural ODE. The first integrals are conserved exactly, which, to my knowledge, previous models do not provide. The first integrals can be learned from data, and the authors show that they correspond to the known conservation, such as conservation of mass and energy. It is notoriously hard to preserve the first integrals of the system in the learned physical simulations, and this work provides an elegant way to do so.

---

> ### Author Response · Authors · 2022-08-01
> **Here is our discussions about the limitations you pointed.**
>
> **Limitation 1-3**
> > The ability of the cFINDE and dFINDE to preserve the first integrals depends on the numerical integrators. dFINDE also depends on the time discretisation delta_t.
>
> > The results in Table 3 suggest that the model is sensitive to K — the number of the first integrals chosen. Moreover, with K>3 the predictions “explode” in case of 2-body in
>
> > FitzHugh–Nagumo systems.
>
> Thank you for pointing these out.
> We agree with you.
> The dFINDE projects the state only at every step and may be sensitive to $\Delta t$.
> In practice, $K$ is a hyperparameter to be adjusted using a validation dataset.
> If there is no validation dataset, $K$ can be set to a modest value (namely, one or two).
>
> In the final version, we will clarify these limitations in Experiments and Conclusion sections.
>
> **Limitation 4**
> > By construction, enforces that certain quantities of the system have to be conserved exactly. The model is not applicable for the systems that are conserving, e.g. having dissipative energy.
>
> Sorry for the confusion, but the FINDE is applicable for systems with dissipative energy.
> The FitzHugh-Nagumo model is just such a case.
> Even then, the model has first integrals (induced by Kirchhoff's laws) other than energy.
> In another example, a pendulum with friction loses its energy, but the length of its rod is preserved.
> The only situation where the FINDE is useless is when the target system does not have any first integrals.
> We will discuss energy-dissipating systems in Section 3 in the final version.

---

> ### Author Response · Authors · 2022-08-01
> **Thank you for insightful comments. Here is our response to your questions.**
>
> We would like to thank you for your valuable comments. We are pleased that you recognized our method as s an elegant way. We would like to address all your concerns below.
>
> **Question 1**
> > Can the authors clarify the difference between the cFINDE and dFINDE?
>
> Sorry for our unclear explanation.
> The main difference is the following: the cFINDE projects the time-derivative at each stage (sub-time step) inside the numerical integrator, whereas dFINDE projects the finite difference at each time step outside the numerical integrator.
> A gradient and discrete gradient are needed for the projection of the time-derivative and finite difference, respectively.
>
> Because the cFINDE performs the projection inside the numerical integrator, first integrals suffer from errors caused by numerical integration.
> In order to eliminate the errors and preserve first integrals exactly, it is necessary to constrain the destination (i.e., finite difference) of the state rather than the direction (i.e., time-derivative).
> For this purpose, we generalize the cFINDE for the finite difference by employing discrete gradients and propose the dFINDE.
>
> We will add the above explanation when introducing the dFINDE in Section 3.2.
>
> **Question 2**
> > In case first integrals are learned, how do you prevent it from learning degenerate solutions, e.g. predicting zero function for all inputs u?
>
> Thank you for your insightful comment.
>
> An algorithm that forces the value of the function $V(u)$ learning first integrals to be zero for a given solution might lead to a constant function.
> However, our algorithm forces its time-derivative $\frac{dV(u)}{dt}$ to be zero, and hence it does not tend to learn a zero function.
>
> Moreover, if the function $V(u)$ becomes a zero function ($V(u)\equiv 0$) or, in general, a constant function, its Jacobian matrix vanishes ($\frac{\partial V(u)}{\partial u}\equiv 0$).
> In this case, our algorithm returns a division-by-zero error because it requires the inverse of the matrix $\frac{\partial V(u)}{\partial u}\frac{\partial V(u)}{\partial u}^\top$ for the projection.
> We have not taken any special measures to prevent such errors, but no errors have occurred in all experiments with proper settings.
> The division-by-zero errors have occurred only when the cFINDE assumes unreasonable many first integrals (e.g., $K=6$ for the double pendulum, which has 5 first integrals).
>
> The FINDE works correctly even when the functions $f(u)$ and $V(u)$ learn the same first integrals; we verified such a case in Section 4.2, where both functions are known.
>
> We will add the above discussions to Experiments section in the final version.
>
> **Question 3**
> > In Figure 1 authors demonstrate that leapfrog integrator does not preserve the total energy, while dFINDE does, while dFINDE uses Dormand–Prince method (dopri5) (mentioned in section 4.1).
>
> Sorry for our confusing presentation.
> Figure 1 provides the results of "1) leapfrog versus dFINDE with leapfrog integrator"; we will clarify this point in the final version.
> In addition, we have added the results of "2) dopri5 versus on dFINDE with dopri5 integrator" to Appendix D.3 in the revised supplementary PDF file.
> Even in this case, the dFINDE preserves the energy more accurately than the base model.
>
> **Question 4**
> > Have authors tested the model on the systems with different masses and total energy than on the training set to verify that the learned constraints indeed learn to compute the total energy/mass that need to be conserved?
>
> Thank you for your important remark.
> Yes, we have tested the model under the situation you pointed out.
>
> We have generated time series from different initial states each time.
> The total energy depends on the state and differs for each time series.
> The total mass of the KdV equation, the rod length of the double pendulum, and the load current of the FitzHugh-Nagumo model also depend on the state and are different for each time series.
> Therefore, we confirmed that the proposed FINDE indeed learns to calculate these first integrals to be conserved.
>
> In the current experimental setting, information about the total mass of the two-body problem is not input to the neural network, and the same value is used for all time series.
> However, given the above experimental results, once the total mass is input, the neural network is expected to learn the relationship between the total mass and first integrals.
>
> We will add the above discussion in the final version.

---

### Official Review · Reviewer_x8Jk · 2022-07-08

**Rating:** 6
**Confidence:** 3
**Soundness:** 3 good
**Presentation:** 4 excellent
**Contribution:** 3 good

**Summary:**

This paper proposes a novel neural model for predicting the dynamics of a system by finding multiple unknown conserved quantities (first integral) from data. The discrete derivative-based formulation eliminates numerical errors, and the model can be trained to strictly satisfy the conservation laws. Experiments on physical systems with multiple conserved quantities show that the proposed model can capture multiple conserved quantities and predict the dynamics of the state more accurately than the base model.

**Questions:**

Q1. When the number K of conserved quantities is unknown, do you have a way to determine K?

Q2. It would be better to specify the objective function and/or the learning algorithm. I was wondering if it is really appropriate to use 1-step error for learning. In [R1,R2], learning is performed using time-window, which might be useful for this study.

Q3. The evaluation experiments are carefully conducted, but I was concerned that the only comparison method is the base model. If possible, it would be better to have a comparison with at least one of the latest models (e.g., DGNet).

Q4. How robust is the proposed model when varying the number of trajectories and time resolution in the training data? It might work well for sparse data because multiple conserved quantities are considered.

[R1] Kevin Course, Trefor Evans, and Prasanth Nair. Weak form generalized Hamiltonian learning. In Advances in Neural Information Processing Systems, volume 33, pages 18716–18726, 2020.

[R2] Yaofeng Desmond Zhong, Biswadip Dey, and Amit Chakraborty. Symplectic ODE-Net: Learn- ing Hamiltonian dynamics with control. In International Conference on Learning Representa- tions, 2020.

**Limitations:**

We believe that learning dynamics via a data-driven approach is effective in the following ways: 1) it allows us to construct simulators even when the mathematical model is unknown, and 2) it can lead to scientific discoveries (e.g., conservation laws). In order to clarify the significance of this research, we would like to ask for clarification of possible applications. In particular, please share your thoughts on how this could lead to scientific discoveries. If there is a gap, it would be good to mention it as a limitation.


**Strengths And Weaknesses:**

- Originality
  - Pros 1: The problem setting of inferring multiple unknown conserved quantities at the same time is new and interesting.
  - Pros 2: Inforporating the prior knowledge of the first integral is reasonable.
  - Pros 3: The formulation based on discrete derivatives is useful.

- Quality
  - Pros 1: This submission is technically sound.
  - Pros 2: Experiments with physical systems with multiple conserved quantities have shown the effectiveness of the proposed model.
  - Cons 1: The description of the learning algorism is somewhat lacking.
  - Cons 2: Only a comparison with the base model is presented.

- Clarity
  - The presentation is excellent.

- Significance
  - Pros 1: Data-driven Approaches to modeling physical systems have attracted considerable attention. It is useful because it has the potential to construct simulators from data for physical phenomena for which mathematical models are unknown.
  - Cons 1: The application is not stated in detail.

---

> ### Author Response · Authors · 2022-08-01
> **Thank you for valuable comments. Here is our replies.**
>
> We would like to thank you for your valuable comments. We are pleased that you appreciate our originality and clarity. We would like to address all your concerns below.
>
> **Limitations**
> > We believe that learning dynamics via a data-driven approach is effective in the following ways: 1) it allows us to construct simulators even when the mathematical model is unknown, and 2) it can lead to scientific discoveries (e.g., conservation laws). In order to clarify the significance of this research, we would like to ask for clarification of possible applications. In particular, please share your thoughts on how this could lead to scientific discoveries. If there is a gap, it would be good to mention it as a limitation.
>
> Thank you for your precious comment.
> We consider that the proposed FINDE contributes more to 2) the scientific discovery.
>
> One of the most famous conservation laws is that associated with Noether's theorem.
> For a system that is derived by the variational principle, if continuous symmetry of a system is found, a corresponding conservation law can be found.
> However, there is no unified way to find general conservation laws.
> FINDE may discover new conservation laws or at least suggest the existence of new conserved quantities.
>
> Many studies have attempted to find the symbolic equations of the underlying dynamics, but the equations are often too complex for symbolic regression to work.
> Even in such cases, the function of the first integral is relatively simple, and symbolic regression can work for it, as shown in Appendix D.1.
>
> Also, the proposed FINDE contributes to 1) constructing simulators.
> In fact, the trained model can be used as a simulator immediately.
>
> In addition, FINDE also has a potential contribution to model reduction.
> If an N-dimensional system has K first integrals, the solution remains an (N-K)-dimensional submanifold. Hence, a (N-K)-dimensional reduced model can be derived, which may significantly reduce computational cost.
> This kind of practical contribution is also expected.
>
> **Question 1**
> > Q1. When the number K of conserved quantities is unknown, do you have a way to determine K?
>
> In practice, $K$ is a hyperparameter to be adjusted using a validation dataset.
> According to Table 3, assuming only one first integral improved prediction performance, and assuming more first integrals tended to improve it further, except for the FitzHugh-Nagumo model.
> If there is no validation dataset, $K$ can be set to a modest value (namely, one or two).
>
> **Question 2**
> > Q2. It would be better to specify the objective function and/or the learning algorithm. I was wondering if it is really appropriate to use 1-step error for learning. In [R1,R2], learning is performed using time-window, which might be useful for this study.
>
> Thank you for your suggestion.
> We agree that a sophisticated learning algorithm is useful for pursuing absolute performance.
> The cFINDE can naturally adopt it, and the dFINDE can adopt it after a minor modification.
> However, such an algorithm requires additional hyperparameters such as the length of prediction time and needs extra effort to adjust them.
> For simplicity and fair comparisons, we used the 1-step error in the present study.
>
> We have added a detailed explanation of the objective functions in Appendix B.3 in the revised supplementary PDF file. Also, we have cited the references you introduced and added the above explanation in that section.
>
> **Question 3**
> > Q3. The evaluation experiments are carefully conducted, but I was concerned that the only comparison method is the base model. If possible, it would be better to have a comparison with at least one of the latest models (e.g., DGNet).
>
> Thank you for your suggestion.
> Models newer than HNN (e.g., DGNet [38], CHNN [20], and dissipative SymODEN [57]) introduced specific prior knowledge about target systems, such as dissipation and constraints.
> In contrast, our proposed FINDE assumes a situation where neural networks learn systems with unknown properties.
> We honestly demonstrated in Table A2 that our proposed method is inferior to the best method (that is, CHNN) when sufficient prior knowledge (that is, holonomic constraints) is available.
>
> **Question 4**
> > Q4. How robust is the proposed model when varying the number of trajectories and time resolution in the training data? It might work well for sparse data because multiple conserved quantities are considered.
>
> Thank you very much for your interesting question.
> As you mention, the FINDE might work better for sparse data.
> However, we do not have sufficient computational resources to demonstrate it in the rebuttal period.
> If it meets the deadline, we will add the results to the final version.

---

> > ### Comment · Reviewer_x8Jk · 2022-08-08
> > **Thank you for your replies.**
> >
> > The additional discussions are also reasonable. The new information resolved my concerns. It would be better to add the above discussion on the contributions of FINDE in the final version.

---

> > > ### Author Response · Authors · 2022-08-08
> > > **Thank you again for your valuable comments.**
> > >
> > > I will add the above discussion to the final version. Your comments will help us to revise our manuscript even better.

---

### Official Review · Reviewer_f3su · 2022-07-11

**Rating:** 5
**Confidence:** 2
**Soundness:** 3 good
**Presentation:** 2 fair
**Contribution:** 2 fair

**Summary:**

This paper proposes to learn the first integrals of an ODE system by a neural network and projects
 the time evolution of the dynamical system onto the submanifold defined using the (discrete) gradients of the learned first integrals. With an appropriate number of assumed first integrals, FINDE predicted future states more accurately than base models.

**Questions:**

1. Can the authors explain more clearly how the implicit dFINDE can be implemented in an explicit way?

2. I don't understand this sentence: "Not only that, FINDE found and preserved the system energy and the total mass as first integrals".  Does it mean that FINDE can't guarantee to preserve other types of first integrals?

**Limitations:**

The limitations have be addressed in the Weaknesses summary.

**Strengths And Weaknesses:**

Strengths:

1. This paper addresses an important topic in predicting an unknown physical system, that is symmetry, or the associated conservation law;

2. The Discrete-Time Projection Method for preserving the first integral of the discretized dynamical system seems to be novel;

3. The empirical results are through and sound.

Weaknesses:

1. The methodology section is hard to follow, no intuition is given after the formal introduction of the method;

2. The contribution of this article is entangled, the authors should separate the question of finding the right first integrals and making more accurate future predictions of the dynamical system. It's hard to say that making a better prediction comes from learning the right first integrals.

---

> ### Author Response · Authors · 2022-07-28
> **Thank you for valuable comments. Here is our response.**
>
> We would like to thank you for your valuable comments. We are glad that you found that "This paper addresses an important topic" and "The empirical results are through and sound." We would like to address all your concerns below.
>
> **Weakness 1**
> > The methodology section is hard to follow, no intuition is given after the formal introduction of the method;
>
> Thank you for your suggestion. In the final version, we will add the following texts to clarify the motivations in Sections 3.1 and 3.2.
>
> (For cFINDE):
> We propose a time-derivative projection method called continuous FINDE (cFINDE) for neural networks.
> Using the method of Lagrange multipliers, the cFINDE projects the time-derivative onto the tangent space ${\mathcal{T}_u\mathcal{M}}'$.
> Roughly speaking, the state movement is bounded in the direction in which the first integral does not change.
> The cFINDE is still an ODE and suffers from errors caused by numerical integration, but it can learn dynamics preserving first integrals $V$, thereby finding unknown first integrals from data.
>
> (For dFINDE):
> In order to eliminate errors caused by numerical integration and to preserve first integrals $V$ exactly, it is necessary to constrain the destination (i.e., finite difference) of the state movement rather than the direction (i.e., time-derivative).
> For this purpose, we generalize the cFINDE for finite difference by employing discrete gradients and propose a projection method called discrete FINDE (dFINDE).
>
> **Weakness 2**
> > The contribution of this article is entangled, the authors should separate the question of finding the right first integrals and making more accurate future predictions of the dynamical system. It's hard to say that making a better prediction comes from learning the right first integrals.
>
> Thank you for your insightful comment.
>
> The main contribution of FINDE is the potential of scientific discovery.
> For a system that is derived by the variational principle, if continuous symmetry of a system is found, a corresponding conservation law can be found by Noether's theorem.
> However, there is no unified way to find general conservation laws.
> FINDE has the potential to discover new conservation laws or at least suggest the existence of new conserved quantities.
>
> Indeed, there is no general theoretical guarantee for the hypothesis that the preservation of first integrals leads to a better prediction or modeling and vice versa.
> However, many previous studies such as HNN, LNN, and DGNet have experimentally supported that the hypothesis holds.
> Our experiments demonstrated that prediction performance tends to increase as more first integrals are found, except for the FitzHugh-Nagumo model, and they supported the hypothesis in a more general way.
>
> In the final version, we will add the above discussions to Background and Conclusion sections.
>
> **Question 1**
> > Can the authors explain more clearly how the implicit dFINDE can be implemented in an explicit way?
>
> Sorry for our unclear explanation.
> For the implicit dFINDE, we defined the state update in Eq. (11);
>
> $$
> \frac{u^{s+1}-u^{s}}{\Delta t^s}=(I-\overline{Y}(u^{s+1},u^{s}))\hat\psi(u^{s};\Delta t^s).
> $$
>
> Given a current state $u^{s}$, the process to obtain the next state $u^{s+1}$ is implicit.
> Therefore, the prediction by the dFINDE is implicit.
>
> However, during the training phase, the ground truth $u_{GT}^{s+1}$ of the next state is known.
> Hence, we assigned the data points $u_{GT}^{s}$ and $u_{GT}^{s+1}$ to both the current state $u^{s}$ and the next state $u^{s+1}$ in Eq. (11).
> Then, the loss function can be the difference between the left- and right-hand sides of Eq. (11), that is,
>
> $$
> ||\frac{u_{GT}^{s+1}-u_{GT}^{s}}{\Delta t^s}-(I-\overline{Y}(u_{GT}^{s+1},u_{GT}^{s}))\hat\psi(u_{GT}^{s};\Delta t^s)||^2_2.
> $$
>
> The discrete Jacobian $\overline{M}$ (and hence $\overline{Y}$) can be obtained explicitly according to Appendix B.2 and the reference [38], and an explicit solver can be used as the numerical integration $\hat\psi$.
> Hence, the process to get the value of the loss function is explicit, and the dFINDE can be trained in an explicit way, whereas its prediction is still in an implicit way.
>
> We have added Appendix B.3 to discuss this point in detail in the revised supplementary PDF file.
>
> **Question 2**
> > I don't understand this sentence: "Not only that, FINDE found and preserved the system energy and the total mass as first integrals". Does it mean that FINDE can't guarantee to preserve other types of first integrals?
>
> Sorry for the confusion caused by my poor writing.
> What we meant is as follows.
>
> FINDE theoretically guarantees to preserve any types of first integrals.
> Our experiments have demonstrated that it conserves many types of first integrals.
> For a complete verification, it would be potential future work to experimentally verify its performance for other types of first integrals.
>
> We will add the above discussions to Conclusion section in the final version.

---

### Author Response · Authors · 2022-08-09
**We are looking forward to hearing from the Reviewers.**

Dear AC,

We are very grateful to the reviewers for their comments, which have helped us to improve the manuscript significantly.

However, we have not yet received a response from Reviewers f3su and kyoV.
Could you please remind the Reviewers to join the discussion with us?

**Reviewer f3su has a confidence rating of 2, the lowest of all reviewers.**
All Weaknesses and Questions pointed out by Reviewer f3su might be due to our unclear explanations, but we have added exhaustive explanations to all of them in our response.
We believe that these additional explanations have enabled Reviewer f3su to evaluate our manuscript with a higher confidence rating.

**Limitation 4 and Question 3 pointed out by Reviewer kyoV are based on misunderstandings.**
We understand that our unclear explanation may have caused these misunderstandings.
However, if possible, we would like Reviewer kyoV to clear these misunderstandings and then re-evaluate our manuscript.

As Limitation 4, Reviewer kyoV wrote:

> By construction, enforces that certain quantities of the system have to be conserved exactly. The model is not applicable for the systems that are conserving, e.g. having dissipative energy.

In fact, the proposed model is applicable for systems having dissipative energy.
In the initial submission, we had already verified this fact using the FitzHugh-Nagumo model, as summarized in Table 2.
Even in this case, the model found first integrals (induced by Kirchhoff's laws) other than energy.

As Question 3, Reviewer kyoV wrote:

> In Figure 1 authors demonstrate that leapfrog integrator does not preserve the total energy, while dFINDE does, while dFINDE uses Dormand–Prince method (dopri5) (mentioned in section 4.1). Presumably, the ability to conserve the integral of dFINDE heavily depends on the integrator used. It is unfair to compare dFINDE equipped with dopri5 versus leapfrog integrator. Can authors provide the results or hypothesise about the following comparisons: 1) leapfrog versus dFINDE with leapfrog integrator 2) dopri5 versus on dFINDE with dopri5 integrator?

In fact, Figure 1 provides the results of "1) leapfrog versus dFINDE with leapfrog integrator"; hence the comparison is already fair.
In addition, we added the results of "2) dopri5 versus on dFINDE with dopri5 integrator" in the revised version.

We are looking forward to hearing from the Reviewers.

Sincerely,

Authors

---

### Meta-Review · Area_Chair_3osU · 2022-08-24

**Recommendation:** Reject
**Confidence:** Certain

**Metareview:**

Although all reviewers find an interesting point and the significance of this paper, some reviewers have several critical concerns in the paper such as the readability (very hard to follow, and lacking some important information) and the non-convincing empirical evaluations. Although it seems that the authors could answer parts of these concerns in their responses and , those seem to require much modification from the original draft. As a total, although this paper could be a good paper after reflecting the reviewers' comments, my recommendation for the current form of this paper is rejection from the relativistic perspective, compared with papers accepted to NeurIPS.

**Award:**

No

---

### Decision · Program_Chairs · 2022-09-14

Reject